# Factors associated with the leftover rate of side dishes in Japanese school lunches

**Kiyo Nakagiri[1], Yukari Sato[2], Takayo Kawakami [3]***

**1** Graduate School of Health and Welfare Science, Okayama Prefectural University, Soja, Okayama, Japan, **2** Department of Contemporary Welfare Science, Faculty of Health and Welfare Science, Okayama Prefectural University, Soja, Okayama, Japan, **3** Department of Nutritional Science, Faculty of Health and Welfare Science, Okayama Prefectural University, Soja, Okayama, Japan

☯ These authors contributed equally to this work.
* kawakami@fhw.oka-pu.ac.jp

**Data Availability Statement:** All relevant data are within the paper and its Supporting Information files.

## Abstract

This study investigated the leftover rate of side dishes in school lunches provided by communal kitchens in Japan's Chugoku region, with a focus on vegetable dishes supplied in containers and three types of menu items served daily to 20 elementary and junior high schools in communal kitchen A for 116 days. First, the leftovers in the containers that were returned to the communal kitchen were weighed and combined. The study then compared outside temperature, distance from communal kitchen A, school type, number of students per class, assignment of nutrition teachers, and time elapsed after cooking. Finally, we examined the relationship between these factors and the leftover rate using multiple regression analysis. The median leftover rate was 20.1% (0–96.9) for 250 side dishes with a high leftover rate; however, this was widely distributed. The number of students per class, assignment of nutrition teachers, and time elapsed after cooking were strongly related to the leftover rate; the adjusted coefficient of determination, $R^2$, was 0.236. The regression results indicated that regarding the side dish leftover rate, the standardized coefficient, β, was 0.414, 0.215, 0.107, 0.093, and 0.094 for the number of students per class, assignment of nutrition teacher, the time elapsed after the end of cooking, distance from communal kitchen A, and presence of seaweed, respectively ($p<0.001$). Dietary education by homeroom and nutrition teachers and reducing the time elapsed after cooking impacts students' awareness and preferences, which may decrease the leftover rate.

## Introduction

Children's health problems are exacerbated by factors such as unbalanced nutritional intake, disordered eating habits, and being overweight or underweight. Studies recommend increasing vegetable and fruit intake in all age groups to decrease the risk of stroke, heart disease, and certain types of cancer [1, 2]. Studies have indicated that the average vegetable intake of the Japanese population is low, especially among those in the age group of 20–40 years [3] and in individuals with low socioeconomic status [4, 5]. In Japan, school lunches are provided in accordance with the "School Lunch Implementation Standards" [6], considering the

**Funding:** The author(s) received no specific funding for this work.

**Competing interests:** The authors have declared that no competing interests exist

nutritional intake necessary for students' growth, individual health, activities of daily living, and local (community) conditions. Therefore, school lunches contribute to healthier eating habits and nutritional intake in students [7]. However, school lunch leftovers remain a challenge in many countries [8–12]; it has been reported that these leftovers do not meet nutritional standards, especially regarding dietary fiber and micronutrients [13]. In Japan, approximately 17.2 kg of food waste is generated per student annually [14], which is a major issue from the perspective of the environment and children's health. Approximately 95.2% of Japanese public and private schools provide lunch, and 41.1% of elementary and 60.2% of junior high schools use communal kitchens, followed by independent kitchens and delivery methods [15]. Although Japanese school lunches are diverse, they comprise a main dish made from meat or fish, a side dish of soup and stew, a second side dish of vegetables, and milk. These menus are uniform across communal kitchens and schools. They are supplied in different types of containers and served to students in classrooms [16, 17]. Communal kitchens often transport various food containers for each class via delivery vehicles for each school, while leftovers from student lunches are returned in containers to the communal kitchen to be weighed and the leftover rate estimated. Although Japan has a nutrition teacher system that manages school lunches and provides food education, these teachers are not present in all schools.

Among studies on school lunch leftovers, researchers have evaluated supplied food items and have discovered that school lunch leftovers were mainly affected by cognitive factors (undesirable food quality and school policies) [18], social factors (meal [eating] time and presence of teacher) [19, 20], and food preferences [12]. Steen et al. [21] studied food waste in schools and pre-school catering units. The authors reported that the serving waste and total waste per portion were mainly related to kitchen type and overproduction rate; however, plate waste was mainly affected by factors related to children's age and stressful eating environments. Izumi et al. [22] examined leftovers from school lunches in Japan through focus interviews with nutrition teachers at elementary schools in Tokyo and revealed that the following factors could help minimize food loss: (1) social norms of eating without leftovers, (2) unfamiliar or uncomfortable foods, (3) nutritional education, (4) portion size and time management, and (5) students' involvement. Wakimoto et al. [23] analyzed leftovers generated in school lunches and reported that the leftover rate in staple foods was associated with room temperature, seasoning, and cooking methods, and milk leftovers were associated with room temperature. Students reportedly dislike eating many vegetables, [24] however, significant portions of side dish leftovers mainly contain vegetables [8], and few studies have examined the factors behind this phenomenon. The School Lunch Hygiene Control Standard [25] stipulates that for the safe and secure provision of school lunches, the temperature of cooked food should be appropriately controlled, and efforts should be made to provide school lunches within two hours of cooking. The delivery process for communal kitchens differs depending on the schools attended; keeping food warm for a long time after cooking may affect children's palatability and the proportion of leftovers. Therefore, schools' educational environment, lunch implementation environment (delivery time and outside temperature), and cooking characteristics (the type of ingredients used in side dishes) may impact leftovers. Several studies on leftover food from school lunches have examined individual or situational factors that influence leftover behavior [12, 19], such as student preferences and attitudes, cafeteria offerings, serving sizes, restricted eating times, and menu factors (such as cooking methods and ingredients used); however, no research has examined the physical and educational environmental factors surrounding schools from a macroscopic perspective. Therefore, in this study, physical environmental factors such as delivery distance, the elapsed time after cooking, and the outside temperature were examined from a novel research viewpoint. Furthermore, we hypothesized

that factors related to the education system, such as the presence of nutrition teachers and class size might affect leftover size. Additionally, we investigated the factors affecting leftover food in schools by surveying and analyzing the measured values of the amount of the leftover food from school lunches delivered from shared kitchens to multiple schools.

Focusing on side dishes, we examined the effects of factors related to the school environment and cooking on the leftover rate of side dishes. We focused on the following environmental factors: outside temperature, distance from the kitchen, school type (elementary or junior high), school size, number of students per class, assignment of nutrition teachers, and the time elapsed from the end of cooking to the commencement of school lunches. Cooking-related factors included seasoning, serving temperature, and ingredients; for example, whether vinegar was used as a seasoning, whether the dish was served hot (side dish with heating only) or cold (side dish with cooling process), or specific inclusion of certain food items (beans, root vegetables, green and yellow vegetables, seaweed).

## Experimental design

### Study area

A total of 2,130 school lunches were distributed to 20 schools, comprising six elementary and 14 junior high schools, in Okayama Prefecture, Japan, via a communal kitchen, A. The percentage of leftovers in the three containers comprising the main dish, soup and stew, and vegetable side dish were measured daily in the communal kitchen A. The weights and percentages of these residuals are listed in S1 Table. We focused on containers of vegetable side dishes supplied to the 20 schools from May 2019 for 116 days (19 [May], 20 [June], 14 [July], 3 [August], 19 [September], 21 [October], and 20 days in November, respectively). S2 Table shows an example of a dish with a large number of vegetable side dishes and their content. Moreover, S3 Table shows the average amount of three cans served and the average amount consumed for two of these representative months. The communal kitchen used a leftover rate exceeding 20% as an indicator of requirement for revising the seasoning and selecting ingredients by conducting menu review meetings; accordingly, when the leftover rate exceeded 20% for a specific side dish in one school, all containers for the side dish were considered for the analysis.

Based on previous studies, we used the quantitative data available from kitchen A for our analysis as risk factors for leftover vegetables. Data on the number of students, number of classes, amount served by dishes, amount of leftovers generated, and outside temperature during the period of lunch supply were obtained from records maintained in the communal kitchen. Communal kitchen A supplied > 10,000 lunches per day and prepared three menu items daily. The three types of menus were, menu A for the five junior high schools, menu B for the six junior high schools, and menu C for the three junior high schools and six elementary schools. The distance of the schools from the communal kitchen varied from approximately 2 to 12 km, resulting in varying times taken to deliver lunches. Therefore, the communal kitchen supplied lunches according to a delivery plan to achieve uniformity in the time elapsed from the end of cooking to the commencement of school lunches among the schools and maintain it under two hours. Accordingly, the delivery distance from the communal kitchen to the school, the end of cooking, and the commencement of school lunch were obtained for each school from the communal kitchen record book. Vegetable side dish ingredients and seasonings were also obtained from records. Nutrition teacher staffing data and temperatures were obtained from official school documents and weather records, respectively.

**Definition of school lunch leftovers and the measurement method.** The measurement of leftovers in side-dish containers was based on a standard method [26]. The communal kitchen A delivered the side dish in containers that were distributed to each class, and the

leftovers were accumulated in separate side dish-specific containers and returned to the kitchen. The returned containers contained the students' leftovers as plate waste and any food not served as serving waste; therefore, in this study, "leftovers" refers to the sum of both.

The person in charge of the communal kitchen weighed the container delivered to each class after the school lunch was ready (for delivery), which was recorded for each container after subtracting the tare weight; this was then combined to calculate the total weight of the food supplied to each school. After lunch, the returned containers were weighed, with the weight recorded without the tare for each container. These were then combined to calculate the total weight of the leftovers for each school.

A digital platform scale (CDP-6700K, Yamato Scale Co., Ltd., Akashi City, Hyogo Prefecture; displaying up to one decimal point) was used to weigh the containers. Cases where containers were not supplied depending on the menu or where the leftover rate was either negative or exceeded 100% were treated as missing values (46 side-dish containers) because of input errors. Finally, we analyzed the data from 1,485 side-dish containers to study leftover rates (S1 Fig).

**Environmental factors as independent variables.**   The independent variables related to the educational environment of schools included school type, school size by the number of students, number of students per class, and the assignment of nutrition teachers. We hypothesized that the leftover rate would increase if the following factors were present: the type of school varied according to the age group of the students, the school size was large, the number of students per class was without any meal adjustment, and nutrition teachers were not assigned. The outside temperature, distance from communal kitchen A, and time elapsed after cooking were considered independent variables. Higher outside temperature may affect children's appetite, and the distance from the kitchen and the time elapsed after completion of cooking may affect school lunch taste and texture; these were considered external physical environment factors that may increase side dish leftovers.

The school types were elementary and junior high schools (elementary school: 0; junior high school: 1). School size was classified according to the number of students in increments of 200 (1, $< 200$; 2, $\geq 200$ to $< 400$; 3, $\geq 400$ to $< 600$; 4, $\geq 600$ to $< 800$; and 5, $\geq 800$). The number of students in each class was calculated by dividing the total number of students by the total number of classes. Nutrition teacher assignments were represented by two categories (assigned school: 0, unassigned school: 1). The outside temperature was measured at 13:00 in communal kitchen A. Communal kitchen A used temperature-controlled food containers (N-cube pots, Nakanishi Mfg. Co. Ltd., https://www.nakanishi.co.jp/product/others/n-cube-pot.html) for delivery via supply vehicles. The distance from communal kitchen A to the respective schools was classified into six categories based on each 2 km increase in radius (1: within a 2 km radius, 2: within 4 km [2–4], 3: within 6 km [4–6], 4: within 8 km [6–8], 5: within 10 km [8–10], and 6: within 12 km [10–12]). When cooking is performed in an *aemono* room (an isolated clean room for adding seasoning to dishes to prevent cross-contamination), seasoning is added to the rotating temperature-maintaining cooking pots while cooling the ingredients. Therefore, the time at which food was transferred from the cooking pot to the side dish containers (for transport) was recorded as the end time of cooking for the line for the day. Accordingly, the elapsed time was calculated as the time elapsed between cooking and school lunch start times.

**Cooking factors.**   Cooking factors were derived from vegetable side dishes that were typically left (S2 Table). Furthermore, the temperature of food delivery, whether hot or cold, was considered another contributing cooking factor. The independent variables included dishes with vinegar as a seasoning, hot or cold dishes, and the inclusion of beans, root vegetables, green and yellow vegetables, and seaweed as ingredients, which were hypothesized to affect children's palatability and increase leftovers.

Two categories were set for using vinegar as seasoning (0: no, 1: yes) and serving temperature (0: cold, 1: hot) of the side dishes in the containers. Hot dishes included simmered or stir-fried dishes prepared using only one heating process, such as simmering, stir-frying, or boiling. Cold dishes included side dishes, such as ingredients with seasonings and salads, in which the vegetables were heated once and subjected to a cooling process. Regarding individual ingredients, beans (soybeans, white flower beans, red kidney beans, etc.) accounting for more than 20% of overall ingredients, root vegetables (radish, lotus root, carrot, burdock, etc.) accounting for more than 30% of all ingredients, green and yellow vegetables (green beans, Japanese mustard spinach, spinach, green peppers, etc.) accounting for more than 20% of the overall ingredients, and seaweeds (wakame, hijiki, etc.) accounting for more than 1 g per person were categorized as binary variables (0: no, 1: yes).

## Data analyses

The Kolmogorov–Smirnov test was used to test the normality of the variables. In addition, a forced-entry multiple regression analysis was performed with the leftover rate of the containers as the dependent variable with environmental and cooking factors as the independent variables. Statistical analyses were performed using IBM SPSS 27.0 for Windows (IBM Corp, Armonk, NY, US) with the significance level set at 5%.

## Ethical consideration

Initial approval and consent were obtained for this study, but in practice, because of the pandemic, interviews and questionnaires were not conducted. Hence, ethical consideration and informed consent are not applicable to this study as the data used are the amount of leftover food from school lunches and environmental data such as temperature collected at schools. These are not human-derived samples, do not contain personal information, and are publicly available data. The study was explained to the Director of Communal Kitchen A and (representative) principals of the schools receiving the lunch supply, and their consent was obtained before conducting the study.

## Results

Table 1 outlines the schools receiving lunch from communal kitchen A and their lunch-delivery statuses. Six elementary and 14 junior high schools received lunch; the median numbers of students in the school and class were 466 and 29.7, respectively. Nutrition teachers were assigned to 4 of the 20 schools: two elementary schools with 400–600 students, one junior high school with 600–800 students, and one junior high school with > 800 students. The average temperature during the study period from early summer to autumn was 25.9˚C ± 5.5˚C. Regarding the distance between communal kitchen A and the receiving schools, five schools were within 2 km, six in the 2–4 km range, four in the 4–6 km range, one in the 8–12 km range, and one in the 10–12 km range.

The mean, median, and maximum leftover rates of side dishes were 21.4, 20.1, and 96.9%, respectively (Table 2). The results of testing for normality using the Kolmogorov–Smirnov test were $p < 0.001$, indicating that normal distribution was not observed.

The school educational environment factors with the side dish leftover rate is shown in Table 3 and the cooking factors with the leftover rate in Table 4.

Next, a forced entry multiple regression analysis was performed with 13 items as independent variables representing factors related to the school environment and cooking: number of students, number of students per class, outside temperature, the time elapsed after the end of cooking, dummy variables used in the categorization of school type, assignment of nutrition

**Table 1. Outline of the schools receiving lunch from communal kitchen A and the school lunch delivery status.**

| School type (number of schools) | Elementary school (6–11 years) | 6 |
|---|---|---|
| | Junior high school (12–14 years) | 14 |
| Number of students (person) | | 466 (357, 656) |
| Number of students per class (person) | | 29.7 (28.1, 31.4) |
| Assignment of nutrition teacher (number of schools) | Yes | 4 |
| | No | 16 |
| Outside temperature (˚C) | | 25.9 ± 5.5 |
| Time elapsed after the end of cooking (Hours) | | 2.3 ± 0.3 |
| Category of distance from communal kitchen to the schools (number of schools) | < 2 km | 5 |
| | 2–4 km | 6 |
| | 4–6 km | 4 |
| | 6–8 km | 3 |
| | 8–10 km | 1 |
| | 10–12 km | 1 |

The number of students and number of students per class are presented as median and interquartile range. The outside temperature and time elapsed after the end of cooking are presented as mean and standard deviation.

teacher, distance from communal kitchen A, seasonings, presence of ingredients (beans, root vegetables, green and yellow vegetables, and seaweed), and serving temperature of side dishes (Table 5).

First, we analyzed the collinearity among the independent variables. School type and the number of students were inversely correlated; accordingly, they were removed from the independent variables. Analysis of variance revealed significant differences between the groups; the adjusted coefficient of determination $R^2$ was 0.236. The variance inflation factor values for the number of students per class, assignment of nutrition teachers, and time elapsed after cooking were 1.19, 1.10, and 1.20, respectively. Next, the regression results showed that, related to the side dish leftover rate, the standardized coefficient β was 0.414, 0.215, 0.107, 0.093, and 0.094 for the number of students per class, assignment of nutrition teacher, the time elapsed after the end of cooking, distance from communal kitchen A, and presence of seaweed, respectively. Thus, increased leftover rate was associated with the presence of seaweed in the side dish, the number of students per class, the time elapsed after the end of cooking, and the distance from communal kitchen A, while the decreased leftover rate was associated with the presence of a nutrition teacher.

**Table 2. Outline of leftover rate of side dishes in containers.**

| | |
|---|---|
| Mean | 21.4 |
| Standard deviation | 14.4 |
| Minimum | 0.0 |
| Maximum | 96.9 |
| 25th percentile | 11.7 |
| 50th percentile | 20.1 |
| 75th percentile | 29.1 |

The values indicate the leftover rate (%) for 250 types of side dishes (1485 instances), where the leftover rate in containers exceeded 20% in at least one school.

**Table 3. Analysis of side dish leftovers in containers by factors related to school environment.**

| | | School lunch delivery instances | Leftover rate (%) |
|---|---|---|---|
| School type | Elementary | 495 | 21.9 (14.4, 30.8) |
| | Junior high | 990 | 13.8 (3.3, 25.1) |
| Number of students | Less than 200 | 162 | 0 (0, 6.6) |
| | 200–less than 400 | 296 | 21.7 (13.6, 31.9) |
| | 400–less than 600 | 476 | 21.4 (12.8, 30.1) |
| | 600–less than 800 | 419 | 21.5 (15.2, 29.4) |
| | More than 800– | 132 | 20.9 (13.7, 31.1) |
| Number of students per class | Less than 30 | 813 | 18.7 (9.8, 28.3) |
| | 30 or more | 672 | 21.8 (13.8, 30.2) |
| Assignment of nutrition teacher | No | 1181 | 21.1 (12.3, 30.6) |
| | Yes | 304 | 16.4 (10.4, 23.6) |
| Distance of communal kitchen | Less than 2 km | 383 | 19 (8.7, 27.6) |
| | 2–4 km | 466 | 19.4 (11.3, 28.7) |
| | 4–6 km | 283 | 20.5 (13.2, 28.7) |
| | 6–8 km | 217 | 20.1 (9.9, 29.3) |
| | 8–10 km | 69 | 20.7 (15.7, 26.6) |
| | 10–12 km | 67 | 31.4 (23.3, 36.7) |
| Outside temperature | Lower than 22.3˚C | 231 | 19.2 (10.6, 26.2) |
| | 22.3˚C–lower than 27.4˚C | 341 | 17.6 (10.6, 26.5) |
| | 27.4˚C–lower than 29.6˚C | 571 | 21.9 (12.7, 34.3) |
| | 29.6˚C or higher | 342 | 20.2 (11.5, 27.7) |
| Time elapsed after end of cooking (hours) | Less than 2 hours | 355 | 15.5 (2.6, 24.6) |
| | 2 hours or more | 1130 | 21.3 (13.1, 30.5) |

The leftover rates are presented in median (25th percentile, 75th percentile)

**Table 4. Cooking factors and side dish leftovers in containers.**

| | | School lunch delivery instances | Leftover rate (%) |
|---|---|---|---|
| Side dish seasoning (vinegar) | Without | 807 | 20.4 (11.7, 29.9) |
| | With | 678 | 19.7 (11.5, 28.4) |
| Side dish (beans) | Without | 1348 | 20.1 (11.3, 29.5) |
| | With | 137 | 20.2 (13.9, 25.9) |
| Side dish (root vegetables) | Without | 1147 | 20.2 (11.8, 29.1) |
| | With | 338 | 19.8 (11.3, 29.2) |
| Side dish (green and yellow vegetables) | Without | 1321 | 20 (11.5, 29.1) |
| | With | 164 | 21.3 (13.7, 29.4) |
| Side dish (seaweed) | Without | 1241 | 19.8 (11.3, 28.7) |
| | With | 244 | 22 (14, 30.7) |
| Side dish (hot or cold) | Hot side dishes (boiled, stir-fried, etc.) | 477 | 21 (11.9, 31) |
| | Cold side dishes (*aemono* [vegetables with seasonings] and salads) | 1008 | 19.7 (11.5, 28) |

The leftover rates are presented as median (25th percentile, 75th percentile).

**Table 5. Investigation of environmental factors, cooking factors, and side dish leftovers (Multiple regression analysis).**

| Independent variables | Unstandardized coefficients | | Standardized coefficients (β) | p-value |
|---|---|---|---|---|
| | B | Standard deviation | | |
| Number of students per class (person) | 0.726 | 0.043 | 0.414 | <0.001 |
| Assignment of nutrition teacher | 7.662 | 0.849 | 0.215 | <0.001 |
| Category of distance from the communal kitchen | 0.979 | 0.244 | 0.093 | <0.001 |
| Outside temperature (˚C) | 0.087 | 0.062 | 0.033 | 0.158 |
| Time elapsed after the end of cooking (hour) | 4.426 | 1.032 | 0.107 | <0.001 |
| Side dish seasoning (vinegar) | 0.087 | 0.823 | 0.003 | 0.916 |
| Side dish (beans) | -3.359 | 1.219 | -0.068 | 0.006 |
| Side dish (root vegetables) | 0.852 | 0.851 | 0.025 | 0.317 |
| Side dish (green and yellow vegetables) | 2.248 | 1.094 | 0.049 | 0.040 |
| Side dish (seaweed) | 3.644 | 1.001 | 0.094 | <0.001 |
| Side dish (hot or cold) | 2.023 | 0.874 | 0.066 | 0.021 |

Multiple regression analysis was performed with side dish leftover rate as a dependent variable and environmental and cooking factors as independent variables. Independent variables were converted into dummy variables; for the environmental variable of assignment of nutrition teacher, "no assignment" was set as 1. For the cooking factor of side dish seasoning "with vinegar," and side dish ingredients containing beans, root vegetables, green and yellow vegetables, and seaweed or not, "presence" was set as 1 and "absence" was set as 0. School type and number of students were excluded from the analysis due to sign reversal.

F = 42.783 ($p < 0.001$). Adjusted coefficient of determination, $R^2$ was 0.236. Durbin–Watson value was 1.397.

## Discussion

Focusing on vegetable side dishes in school lunches in Japan, we investigated the association of the leftover rate with the school educational environment, outside physical environment, and cooking factors. The multiple regression analysis results revealed a significant association between the number of students per class, assignment of nutrition teachers, distance from the kitchen, and time elapsed after cooking. Furthermore, among the cooking factors, green and yellow vegetables and seaweed were significantly associated with the leftover rate.

These results are consistent with some previous studies measuring individual or school-based leftovers (Table 6). Notably, several reports have shown that palatability [19] and food preference [12] affect the amount of leftover plates. Here, we observed that vegetable side dishes tended to remain unconsumed (S2 Table). Therefore, we employed the variables beans, root vegetables, green and yellow vegetables, seaweed, and vinegar, ingredients that children do not like as cooking factors, and compared these with variables examined for their effect on making and serving vegetable side dishes in the present study. We have previously shown that food patterns with Japanese ingredients (dried fish, root vegetables, fish-paste products, and seaweeds) had a higher leftover rate than did those without them [27]. As in the previous study, they tended to leave vegetables with their least favorite tastes. Notably, these foods are becoming increasingly uncommon in home cooking, underscoring the importance of preserving food culture. Ganaha et al. [28] compared eating rates for salads and side dishes featuring eight different seasonings in school lunches, finding no direct impact of seasoning on consumption rates. The authors suggested that the eating rate was likely affected by the dietary environment, such as eating time and homeroom teachers' food education, rather than by seasoning alone. Our results indicated that influence of the presence of foods that children dislike and cooking methods is relatively small. Importantly, palatability among elementary school students has been reported to affect leftovers [12]. In terms of nutrition education, since school lunches endeavor to use local products and Japanese food in the context of food culture, the use of school lunches as educational material throughout the school is deemed important.

**Table 6. Summary of the results of previous studies supporting the educational environment factor, the physical environment factor and the cooking factor, and comparison of these with the variables that were examined for their impact on making and serving vegetable side dishes in the present study.**

| Factor | Effects for plate leftover and food waste of school lunch in previous studies | Reference | Variables | Effect for vegetable side dish leftover in this study |
|---|---|---|---|---|
| Cooking factor | Palatability and preference increase or decrease plate leftover. | [12, 19] | Ingredients that children tend not to like • Beans | decrease |
| | | | • Root vegetable | - |
| | | | • Green and yellow vegetables | increase |
| | | | • Seaweed | increase |
| | | | • Vinegar | - |
| | | | Hot or cold dish | Keep hot increase |
| Physical environment | | | Time elapsed after end of cooking | Long time increase |
| | Satellite kitchen had higher waste level than production unit type kitchens. | [26, 21] | distance from communal kitchen | Long distance increase |
| | Staple foods leftover increase by high room temperature and milk related to low temperature | [23] | Outside temperature | - |
| Educational environment | Children in higher school years produce more plate waste than lower years | [21] | Primary or Junior high as school type | - |
| | Presence of others, dining ambience, noise, increase plate leftover | [21] | Number of students in school | - |
| | Large portion size and short eating time decrease students' intention and increase food waste. | [19, 21, 30, 34] | Number of students per class | increase |
| | Nutrition education for children and teachers reduce food waste | [9, 20] | Nutrition teacher assignment | No assignment of nutrition teacher increase |

Additionally, school lunches are served with the appropriate temperature in mind: hot dishes served hot, and cold dishes served cold. Serving a dish at an appropriate temperature can substantially affect the taste. In the present study, hot vegetable dishes were more frequently left uneaten compared to cold dishes, possibly reflecting children's preferences.

Among the physical environmental factors, Wakimoto et al. [23] reported that the percentage of leftover staple food and milk was affected by temperature; however, in this study, vegetable side dishes were not affected by outside temperature. Notably, the outside temperature was set as a variable that influenced students' appetite. However, during the hot season, the classrooms were air-conditioned and the target cans differed from those in the previous study, which may account for the absence of an observed relationship in the present investigation. On the other hand, Eriksson et al. [26] and Steen et al. [21] reported that kitchen-type satellite systems produce more leftovers than systems where food is prepared in the facility do. Our study hypothesized that the difference in kitchen type could be attributed to physical factors such as delivery distance. Upon examining the length of the delivery route from a satellite unit, we found that greater distances corresponded with an increased number of leftovers. Because the cooking time was obtained from kitchen records, we also focused on the elapsed time after cooking, and it became clear that the longer the time, the more leftovers were found. As mentioned above, temperature control is important in a delivery-type service format for optimal temperature feeding. However, constraints, such as the temperature control of cooked dishes, delivery from communal kitchens to schools, temperature control at schools, equipment in facilities, and the number of cooking workers, remain. Furthermore, depending on the outside temperature, facility equipment, and dishes, providing meals at an appropriate temperature may not always be possible. Currently, communal kitchen A uses temperature-maintaining N-

cube pots (stainless steel double-layer square-shaped containers for food). According to the specifications posted on the official Website, it is highly capable of maintaining the temperature at the serving time. Specifically, Miyoshi et al. [29] note that: "for 8-L of miso soup with pork and vegetables at an initial temperature of 88.1˚C and room temperature of 9.5˚C, kept in a 14-L capacity container, the miso-soup temperature after 2 hours was 75.9˚C; and for 2.3 kg of Japanese mustard spinach with seasonings at an initial temperature of 6.3˚C and room temperature of 26.7˚C, kept in a 7-L capacity container, the temperature of the Japanese mustard spinach after 2 hours was 8.8˚C." Miyoshi et al. evaluated the changes in cooked food due to prolonged maintenance at high temperatures during hot storage and reported that the changes in moisture, hardness, and cohesion increased over time in French fries among the side dishes, while they decreased the sensory evaluation of color, appearance, and texture of bell peppers lightly heated in oil [29]. As identified in this study, the time elapsed after the end of cooking the hot vegetable dishes affected their leftover rate, which may have been due to the change in the quality of cooked food affected by sustained high temperatures. These hot side dishes included sautéed vegetables, stir-fried Japanese mustard spinach with jako (small fish), and stir-fried vegetables. A voluntary appraisal conducted by communal kitchen A with the students in each school and the faculty observed comments such as "stir-fried (and then boiled) Japanese mustard spinach with jako (small fishes) was sticky and the color was not good," and "lightly seasoned and over-juicy." Importantly, as boiled and stir-fried dishes were the main dishes reported regarding the cooking method, it may indicate that they likely led to leftovers, perhaps because of deterioration in texture and color and loss of flavor caused by prolonged temperature maintenance. Derqui et al. [11] investigated food waste in four schools and noted different educational characteristics and models of catering businesses that may differ between cooking in school kitchens and those involving delivery. Kodors et al. [30] attempted to make proposals for decision-making in education and school lunch provision services using simulation methods to consider menu composition and lunchtime with which children could be satisfied. Our study provides evidence that the school lunch delivery system affects leftover food.

Furthermore, studies have found that longer school years, noisy classroom settings [19, 21], and the number of children [21] are risk factors for leftover food. School type and the number of students have also been considered as variables, but the influence was not observed clearly. The fact that this study was conducted mostly in junior high schools may have influenced the differences in the target facilities. In addition to environmental factors within classrooms, several studies have shown that large portion sizes and short eating times decrease students' intentions and increase food waste [9, 19, 21, 30]. From the perspective of the number of students per class, a homeroom teacher can provide detailed instruction when there are fewer students per class. Furthermore, students in charge of lunch will spend less time preparing and cleaning, which will give them enough time to eat, which may help decrease the leftover rate. Martin et al. [9, 20] suggested this association and reported that nutritional educational interventions by teachers led to a decrease in leftovers. It has also been reported that the number of foods proposed affects plate waste [10] and that food waste can be reduced using nudges [31], suggesting that environmental adjustments are important. Moreover, Steen et al. [21] explained that larger portion sizes, possibly due to overproduction, as well as cafeteria noise, contribute to increased leftovers in the public sector, including schools, as factors that affect plate and serving waste by outweighing children's willingness to eat, which leads to leftovers. Our results indicate that schools that did not assign nutrition teachers had more leftover vegetable side dishes than did those. The Ministry of Internal Affairs and Communications' policy evaluation on the promotion of food education reveals that elementary schools with nutrition teachers often spend more time teaching food in each subject than elementary schools without them;

progress has been made in creating a system for engaging in food education throughout the school [32]. Shiori [33] examined the relationship with the assignment of nutrition teachers in the context of environmental factors affecting the awareness and practice of "food education" by teachers of schools offering school lunches. Specifically, the author noted a high level of interest in and a sense of need for "food education" of teachers in charge of classes at compulsory education schools. However, the author observed that without a nutrition teacher, actual awareness and practical activities indicated a lack of progress in the dissemination of food expertise and necessary guidance by teachers working at center schools, where lunches are supplied by a common central kitchen and self-provision schools with their kitchen for school lunches. In the communal kitchen used in this study, a nutrition teacher concomitantly provided food guidance at multiple schools aside from where they were assigned. However, direct instruction was not provided to all classes; instead, broadcast materials incorporating the intent and aim of school lunches were provided. The materials were broadcast during school lunchtime and used for food guidance. Depending on their grade, students may find it difficult to understand the content by merely listening to a broadcast. Rather, if the broadcast information is heard by the class teacher together with the students is used to provide specific guidance based on the actual circumstances of the grade, the school lunch becomes a live teaching material, which increases the effectiveness of food education. Moreover, because adjusting meal distribution reduces leftovers in school lunches [34], we believe that increasing the interest and knowledge of food education among faculty and staff, encouraging them, and building a collaborative system may help reduce leftovers. The impact of cooking factors on the percentage of leftover vegetables may be influenced by the season, time of year, and target population. In addition, in communal kitchens, hot vegetables are delivered and served at the highest possible temperatures in consideration of hygiene and palatability (Fig 1). Our study, based on extensive, long-term school lunch management records, provides new insights into how delivery conditions can lead to increased leftovers, with longer delivery distances and longer periods of maintaining high temperature affecting the quality of the cuisine. The study also presents evidence that the educational environment, such as the classroom environment and the presence of a nutrition teacher, can reduce the risk of leftovers, suggesting that the promotion of nutrition education in schools may reduce the amounts of vegetable leftovers.

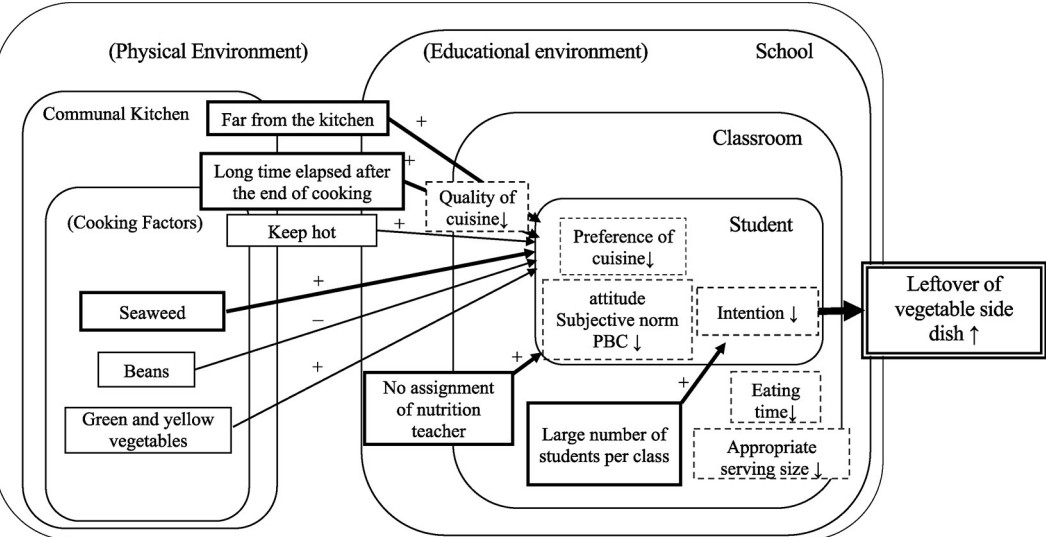

**Fig 1. Summary of factors related to leftover vegetable side dishes in school lunches at communal kitchen A.**

A model is delineated, illustrating the influence of the physical environment, communal kitchen cooking factors, and educational settings on individual student consumption and the subsequent increase in leftover food within the school lunch program. Variables directly pertinent to this study are represented by solid-lined squares. Intra-individual factors, previously elucidated in earlier studies and initially estimated concepts, are indicated by dashed lines.

On the other hand, the study by Donadini et al. [35], which recorded the leftovers of each child and vegetables and fish from the meals of preschool children showed the relationship between the amount of leftovers, preferences and familiarity. Izumi et al. [22] also investigated leftovers in Japanese school lunches through focus interviews with nutrition teachers at elementary schools in Tokyo, and found that the following factors help minimize food loss: (1) social norms for leftover meals, (2) menu planning to increase exposure to unfamiliar and/or disliked foods, (3) nutrition education in school curriculum, (4) teachers' lunch time practice related to portion size and time management, and (5) engagement of student. The social norms that encourages not to leave any leftovers while having meals at home because it is wasteful suggests that parents' eating habits and food choices at home may influence students' food preferences and may lead to a particular pattern of leftover vegetable dishes. Since our study focused only on the percentage of leftover side dishes and examined the results of the school-by-school survey, we were not able to examine the factors involved in detail. However, schools provided daily voluntary inputs regarding vegetable preferences such as "I don't like the taste of fried komatsuna with jako.", "Some people did not like the bitterness of the vegetables.", "The student is uncomfortable with vegetables and seems to have more leftovers than others.", "The radish salad was bitter and pungent." and "If shiitake mushrooms are not a favorite, students seem to leave the entire side dish."

Chu et al. [36] reported on the factors that enable school lunch waste, using a mixed-methodology approach to collect data on school lunches in high schools in Taiwan, combining document analysis, direct weighing, observation, and semi-structured interviews. They identified seven factors contributing to school meal waste: quality of meals, strict budget restrictions, tracking and feedback systems, unanticipated factors, partial meal behaviors, environmental awareness, and lack of initiatives to reduce food waste. In order to reach a better conclusion, further study is needed with the following factors included as those related to leftover vegetables and other side dishes that are considered to have a high percentage of leftover dishes: preferences and selective eating behavior as individual factors, sex ratio as classroom or school factors, unforeseen factors such as absenteeism, food disposal policies and budget limitations as the operating school lunch programs. These factors should be considered in the investigation of food waste management system of school lunch and integrate them in qualitative research methods.

This study had some limitations. First, regarding the adjusted R2, $R^2 \geq 0.5$ was used as a goodness-of-fit measure. However, the corresponding value in this study was low at 0.23, suggesting that we may not have elucidated sufficient factors to explain the leftovers. Unexplained factors may include the short lunchtime of students, preferences, menu composition, stress, and noise in the classroom, which have been described in other studies [18, 19, 21]; these factors were not investigated in this study. Second, as the analysis was conducted with one communal kitchen in a specific municipality, it was limited by constraints unique to the region, limiting the generalizability of our findings. Third, while estimating the number of leftovers, we may have misevaluated the actual number of leftovers, as we did not consider absentees or the difficulty in discerning the edible parts due to limitations in the work of separating the leftovers; this may have resulted in overestimation. Nevertheless, based on the premise of the elementary and junior high schools receiving lunch supplies from a large-scale communal kitchen, we advance the literature by providing empirical evidence on the association between

leftovers and hitherto unexplained factors such as the assignment of nutrition teachers, other environmental factors, and cooking factors.

## Conclusion

The rate of leftover side dishes in school lunches was associated with the number of students per class, the assignment of nutrition teachers, and the time elapsed after cooking. Therefore, food education through lunch guidance by homeroom and nutrition teachers and reducing the time elapsed after cooking can influence students' awareness and preferences, which, in turn, may help reduce leftovers.

## Supporting information

**S1 Fig. Study flow of analytical data.**
(TIF)

**S1 Table. Daily leftovers statistics of side dish containers of primary or jounior high school delivered from communal kitchen A.**
(XLSX)

**S2 Table. Examples of leftover vegetable dishes and content delivered from communal kitchen A.**
(XLSX)

**S3 Table. Leftover rates of side dish containers and nutrition values of primary or jounior high school delivered from communal kitchen A in June and November.**
(XLSX)

**S1 Data. Survey data.**
(XLSX)

## Acknowledgments

We express our deepest gratitude to Minori Nishino, Runa Mikami, Airi Itano, and graduate students Hanai Akiyama, Mari Mitsumori, and Sayaka Yamamoto for their cooperation in data collection. We also express our sincere gratitude to Director Toshifumi Hiramatsu, other staff of Joint Kitchen A in City K, and all others involved in the survey. We would like to thank Editage (www.editage.com) for the English language editing.

## Author Contributions

**Data curation:** Kiyo Nakagiri.

**Formal analysis:** Kiyo Nakagiri, Yukari Sato.

**Investigation:** Kiyo Nakagiri.

**Methodology:** Kiyo Nakagiri.

**Project administration:** Takayo Kawakami.

**Supervision:** Takayo Kawakami.

**Validation:** Yukari Sato.

**Writing – original draft:** Kiyo Nakagiri.

**Writing – review & editing:** Yukari Sato, Takayo Kawakami.

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
