## [Decision Letter · Decision Letter 0]

15 Aug 2023

PONE-D-23-10152Factors associated with the leftover rate of side dishes in Japanese school lunchesPLOS ONE

Dear Dr. Kawakami,

Thank you for submitting your manuscript to PLOS ONE. After careful consideration, we feel that it has merit but does not fully meet PLOS ONE’s publication criteria as it currently stands. Therefore, we invite you to submit a revised version of the manuscript that addresses the points raised during the review process.

ACADEMIC EDITOR: Please kindly attend to comments below, as well as attached.

We look forward to receiving your revised manuscript.

Kind regards,

Charles Odilichukwu R. Okpala

Academic Editor

PLOS ONE

Journal Requirements:

2. You indicated that ethical approval was not necessary for your study. We understand that the framework for ethical oversight requirements for studies of this type may differ depending on the setting and we would appreciate some further clarification regarding your research. Could you please provide further details on why your study is exempt from the need for approval and confirmation from your institutional review board or research ethics committee (e.g., in the form of a letter or email correspondence) that ethics review was not necessary for this study? Please include a copy of the correspondence as an ""Other"" file

3. Please amend your current ethics statement to address the following concerns:

a) Did participants provide their written or verbal informed consent to participate in this study?

4. We noted in your submission details that a portion of your manuscript may have been presented or published elsewhere. [Yes, I recently reported using the same data source in the Japanese article below.

A trial of food pattern analysis and leftover food evaluation of side dishes in a school lunch central kitchens in Okayama Prefecture. 7(1)72-80.2023,Bulletin of Higher Education and Liberal Arts and Sciences Research, Okayama Prefectural University doi/10.15009/00002459

However, in this paper, principal component analysis was performed from the weight of 17 food groups used in all canned side dishes, not just canned vegetables, to summarize the characteristics of school lunch side dish menus. The difference between this paper and the paper in preparation is the purpose (relationship of different factors such as environment, education, and cooking factors related to leftovers, targeting canned vegetables with a high rate of leftovers/establishment of a simple menu evaluation method). The method (principal component analysis/multiple regression analysis) is different. Two main factors were identified: ``Japanese ingredients'' (dried fish, root vegetables, seafood paste products) and ``main dish ingredients'' (fish and shellfish, meat). The result that the rate of leftover food is high in menus that use Japanese ingredients and/or fish as the main ingredient is different from the result of this paper.] Please clarify whether this [conference proceeding or publication] was peer-reviewed and formally published. If this work was previously peer-reviewed and published, in the cover letter please provide the reason that this work does not constitute dual publication and should be included in the current manuscript

Additional Editor Comments :

Please authors kindly address in very great detail all comments raised ok. Thank you

Reviewers' comments:

Reviewer's Responses to Questions

**Comments to the Author**

1. Is the manuscript technically sound, and do the data support the conclusions?

Reviewer #1: Partly

Reviewer #2: Partly

2. Has the statistical analysis been performed appropriately and rigorously? 

Reviewer #1: Yes

Reviewer #2: Yes

3. Have the authors made all data underlying the findings in their manuscript fully available?

Reviewer #1: Yes

Reviewer #2: Yes

4. Is the manuscript presented in an intelligible fashion and written in standard English?

Reviewer #1: Yes

Reviewer #2: Yes

5. Review Comments to the Author

Reviewer #1: Decision: Major Revision.

Study of leftover food is the one of the best research area, because food waste is one of the major problem in the world in the point of economic, social and environmental. But, according to the following comments you should be corrected and revised the manuscripts.

Comment

It is preferable to substitute "the study or the research" in instead of "we" in lines 26 and 29 of the abstract section on page 2.

Under methodology part, line 113-114 what is your preference for selection of school/elementary?

What are your preferences for choosing dates and months under the methodology component, lines 114–116? Why not choose the other months of the year? Also, why choose more dates for some months while doing the opposite for others. Why not use the same date, for instance, 3 for August and 21 for October?

Please identify the standard reference for the formula used to compute the total weight of the leftovers under analysis items, lines 145–146. I believe it is not necessary to write the definition of leftover food under this heading, hence the caption "Definition of school lunch leftovers and the measurement method" needs to be changed.

It is better to use "study area" or "sit of study" in place of "participant" in the methodology section. Additionally, it is preferable to use the caption "experimental design" to write the experimental design and procedure in a single paragraph.

it is preferable to incorporate the information mentioned under "environmental factor" and "cooking factor" (independent variables) under "experimental design".

Page 9, line 202, the caption "analysis method" should be changed by "data analysis method" because you were analyzing the experimental data under this.

The table arrangement in your result part is unclear, therefore please rewrite the table in smart format.

You ought to discuss the data in the table in relation to fact science (by citing the appropriate references) and compare it to earlier research that is related to this work. It is preferable to present the results in a table and discuss them at the same time.

You must illustrate the impact of each independent component in the results section, including how they affected the amount of leftover food, both verbally and graphically. What happens when it comes to the leftover food for students as an independent element changes (increases and decreases).

Generally, under this work

There is weak methodology (the experimental setup/design of the research, the methodologies for analysis, etc. are not written clearly).

There is insufficient result discussion (the results you include in the table must be bolstered by an in-depth explanation, parallels, and references...)

Lack of explanation or discussion of the impact of independent elements (cooking and environmental conditions) on the student's meal and how those factors affect the amount of leftover food?

Reviewer #2: I thank the authors for their efforts in writing this paper. Generally, it reads well but I have a couple of issues to be addressed in this paper. These are:

1. The relevance of the manuscript must be clearly discussed- how does the factors associated with leftover rate of side dishes contributes to the nutrition intake of the school lunches. I perceived this article is part of a bigger research that was conducted and presenting the findings alone without the other parts of the work makes it incomplete. What exactly is the new knowledge that this manuscript is contributing to the existing knowledge in this field. Add a section on the new knowledge that this paper contributes to its audience- more like using an expository approach to dig deep into how these factors identified could improve the school lunch programs.

2. Why focus on the leftover of the side dishes and not the main dishes? Is there an important message that the authors would like to convey to their readers about side dishes? The context of side dishes and its contribution to the overall nutrient intake of the population should be well articulated in the background section. The methodology of the manuscript should be revised to include what these side dishes, part of the side dishes that were always left over by the subjects and many other information that could provide the reader some context.

3. Interested to know the content of the leftover and how much of these contributed to the food waste in the schools.

4. Add a section on the implication of the findings of this study.

5.The results section seems to have lengthy description of the tables - I suggest only relevant data should be described. The discussion of the results was inadequately completed- more work should be done to bring out clarity about the results and the relevance of this study.

6. PLOS authors have the option to publish the peer review history of their article (what does this mean?). If published, this will include your full peer review and any attached files.

Reviewer #1: No

Reviewer #2: No

---

## [Author Response · Author response to Decision Letter 0]

12 Oct 2023

PONE-D-23-10152

Factors associated with the leftover rate of side dishes in Japanese school lunches

We would like to thank the Editor and Reviewers for their constructive critique to improve the manuscript. We have made every effort to address the issues raised and to respond to all comments. Please, find next a detailed, point-by-point response to the comments. 

Response to Editor

2. You indicated that ethical approval was not necessary for your study. We understand that the framework for ethical oversight requirements for studies of this type may differ depending on the setting and we would appreciate some further clarification regarding your research. Could you please provide further details on why your study is exempt from the need for approval and confirmation from your institutional review board or research ethics committee (e.g., in the form of a letter or email correspondence) that ethics review was not necessary for this study? Please include a copy of the correspondence as an ""Other"" file.

Response: According to the guidelines provided by the Research Ethics Committee of Okayama Prefectural University, ethics approval is not required for research projects that meet specific conditions. Firstly, if a project does not involve human subjects or data, materials of human origin, it is exempt from requiring an ethics application. Secondly, even if the project involves human subjects or identifiable data, it is still exempt if the information and data meet both of the following conditions:

- Data are existing and publicly available

- The subjects cannot be identified by any means

Please find attached the email exchange we had with the Ethics Committee Secretariat.

3. Please amend your current ethics statement to address the following concerns:

a) Did participants provide their written or verbal informed consent to participate in this study?

Response: No, they did not. 

Response: Not applicable.

4. We noted in your submission details that a portion of your manuscript may have been presented or published elsewhere. [Yes, I recently reported using the same data source in the Japanese article below.

A trial of food pattern analysis and leftover food evaluation of side dishes in a school lunch central kitchens in Okayama Prefecture. 7(1)72-80.2023,Bulletin of Higher Education and Liberal Arts and Sciences Research, Okayama Prefectural University doi/10.15009/00002459

However, in this paper, principal component analysis was performed from the weight of 17 food groups used in all canned side dishes, not just canned vegetables, to summarize the characteristics of school lunch side dish menus. The difference between this paper and the paper in preparation is the purpose (relationship of different factors such as environment, education, and cooking factors related to leftovers, targeting canned vegetables with a high rate of leftovers/establishment of a simple menu evaluation method). The method (principal component analysis/multiple regression analysis) is different. Two main factors were identified: ``Japanese ingredients'' (dried fish, root vegetables, seafood paste products) and ``main dish ingredients'' (fish and shellfish, meat). The result that the rate of leftover food is high in menus that use Japanese ingredients and/or fish as the main ingredient is different from the result of this paper.] 

Please clarify whether this [conference proceeding or publication] was peer-reviewed and formally published. If this work was previously peer-reviewed and published, in the cover letter please provide the reason that this work does not constitute dual publication and should be included in the current manuscript

Response: As you pointed out, this paper diverges from the previous report in its primary objectives, methodology, and subject selection, as well as its outcomes. This manuscript is intended for a university-published bulletin and, as per the submission guidelines, is subject to peer review. In my cover letter, I articulated the reasons for including this study in the current manuscript, emphasizing that it does not amount to double publication.

Response: There are no ethical or legal restrictions. We request that our data be uploaded in the form of minimally anonymized data necessary to reproduce the results of our study.

Response: SupportingInformation_suppl_data_23104 as a Supporting Information file.

 

Reviewer #1: Decision: Major Revision.

Thank you very much for your advice.

Response to Comment

It is preferable to substitute "the study or the research" in instead of "we" in lines 26 and 29 of the abstract section on page 2.

Response: Thank you for your insightful review. I have made the correction as you suggested. 

Under methodology part, line 113-114 what is your preference for selection of school/elementary?

Response: We appreciate the reviewer's attention to detail. All schools served by communal Kitchen A were included in this study. No exclusions were made, and no specific selection criteria were applied for the choice of schools. Although the city of A contains other elementary and junior high schools with independent kitchens, we focused exclusively on schools served by Kitchen A. This approach enabled us to investigate the impact of food preparation, environmental factors, and educational approaches on leftover food, all based on side dishes originating from the same kitchen. It should be noted, however, that one limitation of our sampling is that fewer elementary schools than junior high schools were included among the target schools. This is primarily because, in the city of A, a larger percentage of elementary schools have their own kitchens.

What are your preferences for choosing dates and months under the methodology component, lines 114–116? Why not choose the other months of the year? Also, why choose more dates for some months while doing the opposite for others. Why not use the same date, for instance, 3 for August and 21 for October?

Please identify the standard reference for the formula used to compute the total weight of the leftovers under analysis items, lines 145–146. I believe it is not necessary to write the definition of leftover food under this heading, hence the caption "Definition of school lunch leftovers and the measurement method" needs to be changed.

Response: We appreciate the reviewer's attention to detail. To clarify, our analysis encompassed all school days in the months of May, June, July, August, September, October, and November. The aim was to examine the environmental factors affecting school lunch leftovers during periods that include both the hottest and most temperate times of the year, when maximum temperatures are at their peak. The number of school lunch days varies monthly, as lunches are not provided during school events and holidays. Additionally, there are fewer school lunches in August in Japan due to the extended vacation period.

It is better to use "study area" or "sit of study" in place of "participant" in the methodology section. Additionally, it is preferable to use the caption "experimental design" to write the experimental design and procedure in a single paragraph.

Response: Thank you for these valuable suggestions. We have incorporated the official standard references and amended the section to "The Measurement Method of School Lunch Leftovers." 

it is preferable to incorporate the information mentioned under "environmental factor" and "cooking factor" (independent variables) under "experimental design".

Page 9, line 202, the caption "analysis method" should be changed by "data analysis method" because you were analyzing the experimental data under this.

Response: Thank you for your valuable insights. I have opted for the caption "study area" in place of "participant," and have also utilized "experimental design" for further clarity. 

The table arrangement in your result part is unclear, therefore please rewrite the table in smart format.

Response: Thank you for your comment. We have revised the arrangement of the tables accordingly.

You ought to discuss the data in the table in relation to fact science (by citing the appropriate references) and compare it to earlier research that is related to this work. It is preferable to present the results in a table and discuss them at the same time.

Response: Thank you to the reviewer for the valuable feedback. I have displayed the findings in Table 6 and included an additional explanation in the Discussion section. 

You must illustrate the impact of each independent component in the results section, including how they affected the amount of leftover food, both verbally and graphically. What happens when it comes to the leftover food for students as an independent element changes (increases and decreases).

Response: Thank you for the constructive suggestion. I have incorporated an illustrated Figure 1 into the Results section and have also added a corresponding explanation to the Discussion. 

Generally, under this work

There is weak methodology (the experimental setup/design of the research, the methodologies for analysis, etc. are not written clearly).

Response: Thank you for your valuable advice. I have incorporated the experimental setup, study design, and analytical methods into the Methods section of the study.

There is insufficient result discussion (the results you include in the table must be bolstered by an in-depth explanation, parallels, and references...)

Response: Thank you for the insightful suggestion. I have included a table to elucidate the congruence between the results of this study and those of prior research, thereby highlighting the significance of the present investigation. 

Lack of explanation or discussion of the impact of independent elements (cooking and environmental conditions) on the student's meal and how those factors affect the amount of leftover food?

Response: Thank you for the valuable suggestion. I have included an illustration to elucidate the influence of cooking and environmental factors on the leftovers of side dishes in school lunches. 

 

Reviewer #2:

Thank you very much for your advice.

Response to Comment 

1. The relevance of the manuscript must be clearly discussed- how does the factors associated with leftover rate of side dishes contributes to the nutrition intake of the school lunches. I perceived this article is part of a bigger research that was conducted and presenting the findings alone without the other parts of the work makes it incomplete. What exactly is the new knowledge that this manuscript is contributing to the existing knowledge in this field. Add a section on the new knowledge that this paper contributes to its audience- more like using an expository approach to dig deep into how these factors identified could improve the school lunch programs.

Response: We appreciate the reviewer's attention to detail. In this study, we collected data on the percentage of leftover vegetables in three different containers, as well as the types of ingredients used in each side dish, from this communal kitchen. However, the analysis only incorporated data on the leftover percentage of vegetables from May to November and data on the types of ingredients, seasoning, and cooking temperatures used in side dishes. Indeed, a more comprehensive approach would have included an assessment of how these leftover side dishes contributed to the nutrient content of the menu items served in the elementary and junior high schools associated with this kitchen. Therefore, we utilized available data from 40 representative days in June and November to ascertain the nutritional composition of the entire menu or individual side dishes, as well as the percentage of leftover canned vegetables in main dishes, side dishes, soups, and stews. These statistics are compiled in the accompanying tables. According to Table S-1, looking at the nutrient intake of side dishes after subtracting the leftover vegetables, the intake of dietary fiber, calcium, and iron, which are difficult to obtain even in daily home meals, is low even in school lunches, and the contribution of vegetable side dishes and soup leftovers is considered to be significant. As this does not constitute a complete data analysis, it limits our ability to discuss the study's impact comprehensively. Nonetheless, these figures serve as representative values for the warmer and more temperate months and are included for reference.

In addition, the new knowledge that this paper contributes to existing knowledge is highlighted around line 425-430.

2. Why focus on the leftover of the side dishes and not the main dishes? Is there an important message that the authors would like to convey to their readers about side dishes? The context of side dishes and its contribution to the overall nutrient intake of the population should be well articulated in the background section. The methodology of the manuscript should be revised to include what these side dishes, part of the side dishes that were always left over by the subjects and many other information that could provide the reader some context.

Response: Thank you for your review. The significance of vegetable intake, coupled with the fact that the nation's overall vegetable consumption is low—particularly among lower socioeconomic households—was highlighted in the background. Furthermore, it was emphasized that the inclusion of vegetable side dishes in school lunches helps to mitigate this disparity in children's diets.

Statistics pertaining to the leftovers of side dishes—as well as main dishes and soups or stews—in school lunches are provided in a supplemental table (SupportingInformation_fileS1). Additionally, a detailed account of the quantity and content of leftover side dishes has been incorporated into the Methods section. We extend our gratitude to the reviewer for highlighting the importance of these aspects.

3. Interested to know the content of the leftover and how much of these contributed to the food waste in the schools.

As shown in Supplemental table 1, food waste at schools is approximately 3.2 kg of main dishes, 11.8 kg of soup, and 6.0 kg of side dishes per day, with side dishes accounting for 28.5% of the remaining amount. Of these, soups include vegetables, but also include protein-based foods, meat and meat products and soup, and because of their high serving weight, they also have the highest weight of leftover vegetables. However, side dishes have the highest percentage of leftover vegetables, although they are served with less weight than soups and stews, and thus are considered to be the most representative of the tendency of leftover vegetable dishes. The main types and contents of vegetable side dishes that are often left and uneaten the content of the leftover shown in Supplemental table 2.

4. Add a section on the implication of the findings of this study.

Response: We have added the section to the discussion (pages 25, lines 425-430). We appreciate the reviewer's insightful comment.

5.The results section seems to have lengthy description of the tables - I suggest only relevant data should be described. The discussion of the results was inadequately completed- more work should be done to bring out clarity about the results and the relevance of this study.

Response: Thank you for your valuable advice. I have streamlined the description of the results to focus solely on pertinent data, and have also incorporated figures and tables to elucidate both the results and the study's relevance.

---

## [Decision Letter · Decision Letter 1]

20 Nov 2023

PONE-D-23-10152R1Factors associated with the leftover rate of side dishes in Japanese school lunchesPLOS ONE

Dear Dr. Kawakami,

Thank you for submitting your manuscript to PLOS ONE. After careful consideration, we feel that it has merit but does not fully meet PLOS ONE’s publication criteria as it currently stands. Therefore, we invite you to submit a revised version of the manuscript that addresses the points raised during the review process.

**ACADEMIC EDITOR: Please see comments below**

We look forward to receiving your revised manuscript.

Kind regards,

Charles Odilichukwu R. Okpala

Academic Editor

PLOS ONE

Journal Requirements:

Additional Editor Comments (if provided):

Please, authors, kindly address concerns raised by a new invited reviewer. It will help improve the quality of this work. Thank you very much

Reviewers' comments:

Reviewer's Responses to Questions

**Comments to the Author**

1. If the authors have adequately addressed your comments raised in a previous round of review and you feel that this manuscript is now acceptable for publication, you may indicate that here to bypass the “Comments to the Author” section, enter your conflict of interest statement in the “Confidential to Editor” section, and submit your "Accept" recommendation.

Reviewer #2: All comments have been addressed

Reviewer #3: (No Response)

2. Is the manuscript technically sound, and do the data support the conclusions?

Reviewer #2: Yes

Reviewer #3: Partly

3. Has the statistical analysis been performed appropriately and rigorously? 

Reviewer #2: Yes

Reviewer #3: N/A

4. Have the authors made all data underlying the findings in their manuscript fully available?

Reviewer #2: Yes

Reviewer #3: Yes

5. Is the manuscript presented in an intelligible fashion and written in standard English?

Reviewer #2: Yes

Reviewer #3: Yes

6. Review Comments to the Author

Reviewer #2: I thank the authors for sufficiently addressing the questions- the paper is now richer in content and clearly fits well with modern science.

Reviewer #3: Thank you for your valuable work. The study is interesting and you can see my suggestions below;

1- The School Lunch Implementation Standards may provide details for the ingredients of students’ meals however, they cannot persuade students to eat all meals on their dishes. Here we can talk about many different factors such as the eating habits of their parents, the eating habits of students or food preferences of students. So how did you evaluate these characteristics?

2- To decide what are the main reasons for leftover rates/types of students these (school educational environment, outside physical environment, and cooking factors) cannot be the only factors. There are many other factors and using only defined factors is not enough.

3- You present the results with statistical evaluations, however, evaluated factors are not enough to obtain a final conclusion about the leftovers of students in junior and high schools. For palatabilities of children/adolescents, there are many studies with varied evaluation criteria, however, your criteria are not enough to get a final decision. You should broaden your criteria to reach a better conclusion for the study.

4- l think it will be better if you read these articles below and improve your discussion part as well;

*doi.org/10.3390/su15075947

*doi.org/10.1016/j.ijgfs.2022.100520

7. PLOS authors have the option to publish the peer review history of their article (what does this mean?). If published, this will include your full peer review and any attached files.

Reviewer #2: **Yes: **Dr. Paul Eme

Reviewer #3: No

---

## [Author Response · Author response to Decision Letter 1]

26 Jan 2024

PONE-D-23-10152

Factors associated with the leftover rate of side dishes in Japanese school lunches

We would like to thank the Editor and Reviewers for their constructive critique to improve the manuscript. We have made every effort to address the issues raised and to respond to all comments and suggestions. The revisions are indicated with red font in the revised manuscript. Please, find below a detailed, point-by-point response to the Editor’s and Reviewers’ comments, and we hope that our revisions will meet the Editor’s and Reviewers’ expectations.

Point-by-point responses to Editor’s and Reviewers’ comments

Response to Editor’s comments:

Journal Requirements:

Response: Thank you for appraising us about the journal policies regarding Reference list. We have reviewed all references and replaced the URL of reference No. 15 with the most recent one (pasted below for your reference).

“15. Survey on school lunch implementation status. Ministry of Education, Culture, Sports, Science, and Technology; February 2019(in Japanese) [cited Dec 12, 2023]. Available from: https://warp.ndl.go.jp/info:ndljp/pid/11293659/www.mext.go.jp/b_menu/toukei/chousa05/kyuushoku/kekka/k_detail/__icsFiles/afieldfile/2019/02/26/1413836_001_001.pdf. (Heisei 30).”

Response to Reviewers' comments:

Reviewer #2: I thank the authors for sufficiently addressing the questions- the paper is now richer in content and clearly fits well with modern science.

Response: We greatly appreciate your valuable comment and assessment of our work, Dr. Paul Eme.

Reviewer #3: Thank you for your valuable work. The study is interesting and you can see my suggestions below;

1- The School Lunch Implementation Standards may provide details for the ingredients of students’ meals however, they cannot persuade students to eat all meals on their dishes. Here we can talk about many different factors such as the eating habits of their parents, the eating habits of students or food preferences of students. So how did you evaluate these characteristics?

Response: Thank you very much for your valuable comments and for introducing the papers. Certainly, there are many factors that may affect consumption of school lunches, including parents' eating habits, students' eating habits, and students' food preferences. As per your suggestion, we have elaborated on this aspect in our revised manuscript as follows: 

In lines 445-462, we have mentioned the feedback comments from students and teachers on the days when there was most leftover vegetables which were considered in association with the warming time, but not to any specific preferences for the vegetables themselves. We have added the following text to improve the discussion section.

“On the other hand, the study by Donadini et al. [35], which recorded the leftovers of each child and vegetables and fish from the meals of preschool children showed the relationship between the amount of leftovers, preferences and familiarity. Izumi et al. [22] also investigated leftovers in Japanese school lunches through focus interviews with nutrition teachers at elementary schools in Tokyo, and found that the following factors help minimize food loss: (1) social norms for leftover meals, (2) menu planning to increase exposure to unfamiliar and/or disliked foods, (3) nutrition education in school curriculum, (4) teachers’ lunch time practice related to portion size and time management, and (5) engagement of student. The social norms that encourages not to leave any leftovers while having meals at home because it is wasteful suggests that parents' eating habits and food choices at home may influence students' food preferences and may lead to a particular pattern of leftover vegetable dishes. Since our study focused only on the percentage of leftover side dishes and examined the results of the school-by-school survey, we were not able to examine the factors involved in detail. However, schools provided daily voluntary inputs regarding vegetable preferences such as “I don't like the taste of fried komatsuna with jako.”, “Some people did not like the bitterness of the vegetables.”, “The student is uncomfortable with vegetables and seems to have more leftovers than others.”, “The radish salad was bitter and pungent." and “If shiitake mushrooms are not a favorite, students seem to leave the entire side dish.””

2- To decide what are the main reasons for leftover rates/types of students these (school educational environment, outside physical environment, and cooking factors) cannot be the only factors. There are many other factors and using only defined factors is not enough.

Response: Thank you for this comment. We have added the observations of Chu et al. (doi: 10.3390/su15075947) where they have analyzed the causal factors that result in school lunch waste, using a mixed-methodology approach to collect data on school lunches in high schools in Taiwan. They identified seven factors contributing to school meal waste. 

Lines 463-468: “Chu et al. [36] reported on the factors that enable school lunch waste, using a mixed-methodology approach to collect data on school lunches in high schools in Taiwan, combining document analysis, direct weighing, observation, and semi-structured interviews.　They identified seven factors contributing to school meal waste: quality of meals, strict budget restrictions, tracking and feedback systems, unanticipated factors, partial meal behaviors, environmental awareness, and lack of initiatives to reduce food waste.”

We believe that the above factors may also be contributing to leftover school lunches, which in turn are contributing to school-wide food waste.

3- You present the results with statistical evaluations, however, evaluated factors are not enough to obtain a final conclusion about the leftovers of students in junior and high schools. For palatabilities of children/adolescents, there are many studies with varied evaluation criteria, however, your criteria are not enough to get a final decision. You should broaden your criteria to reach a better conclusion for the study.

Response: Thank you very much for your insightful comment. This study was conducted based on the measurement records of the ratio of leftover side dishes in each school, and did not examine factors among the consumers or operational factors that reduce the ratio of leftover vegetables, which we believe is insufficient as you have pointed out. We have added the following text to elaborate this point as follows:

Lines 468-474: “In order to reach a better conclusion, further study is needed with the following factors included as those related to leftover vegetables and other side dishes that are considered to have a high percentage of leftover dishes: preferences and selective eating behavior as individual factors, sex ratio as classroom or school factors, unforeseen factors such as absenteeism, food disposal policies and budget limitations as the operating school lunch programs. Future considerations include inclusion in the requirements for factor investigation of food waste management system of school lunch and integration with qualitative research methods.

4- l think it will be better if you read these articles below and improve your discussion part as well;

*doi.org/10.3390/su15075947

*doi.org/10.1016/j.ijgfs.2022.100520

Response: Thank you very much for this suggestion. We have included the above papers in our reference list together with comments in the discussion.

“35. Donadini G, Spigno G, Fumi MD, Porretta S. School lunch acceptance in pre-schoolers. Liking of meals, individual meal components and quantification of leftovers for vegetable and fish dishes in a real eating situation in Italy. Int J Gastronomy Food Sci. 2022;28:100520. doi: 10.1016/j.ijgfs.2022.100520.

36. Chu C, Chih C, Teng C. Food waste management: A case of Taiwanese high school food catering service. Sustainability. 2023;15(7):5947. doi: 10.3390/su15075947.”

In addition to the above comments, all spelling and grammatical errors have been corrected. 

We look forward to hearing from you in due time regarding our submission and to respond to any further questions and comments you may have.

Once again, we thank you very much for the comments and suggestions.

---

## [Editor Report · Decision Letter 2]

30 Jan 2024

Factors associated with the leftover rate of side dishes in Japanese school lunches

PONE-D-23-10152R2

Dear Dr. Kawakami,

We’re pleased to inform you that your manuscript has been judged scientifically suitable for publication and will be formally accepted for publication once it meets all outstanding technical requirements.

Kind regards,

Charles Odilichukwu R. Okpala

Academic Editor

PLOS ONE

Additional Editor Comments (optional):

Thank you for revising your manuscript. It is now acceptable for publication.

Thank you for finding PlosONE as your journal of choice. Look forward to your future scholarly contributions

Congratulations
---

## [Editor Report · Acceptance letter]

14 Feb 2024

PONE-D-23-10152R2 

PLOS ONE

Dear Dr. Kawakami, 

I'm pleased to inform you that your manuscript has been deemed suitable for publication in PLOS ONE. Congratulations! Your manuscript is now being handed over to our production team.

Kind regards, 

on behalf of

Dr. Charles Odilichukwu R. Okpala 

Academic Editor

PLOS ONE